# Health Outcomes and Resource Consumption Analysis of Radioembolization with Y90 Glass Microspheres (TARE-Y90) Versus Transarterial Chemoembolization with Irinotecan (DEBIRI) in Patients with Liver Metastases from Colorectal Cancer in Spain

**DOI:** 10.3390/diagnostics15070796

**Published:** 2025-03-21

**Authors:** Juan José Ciampi-Dopazo, Gonzalo Ruiz Villaverde, Juan José Espejo, Raúl García Marcos, Daniel Pérez Enguix, Serena Pisoni, José J. Martínez-Rodrigo, Pablo Navarro Vergara, Pedro Pardo Moreno, Antonio Rodríguez-Fernández

**Affiliations:** 1Interventional Radiology Unit, Hospital Universitario Virgen de las Nieves, 18014 Granada, Spain; lalousso@hotmail.com (G.R.V.); pablofnavarro@gmail.com (P.N.V.); pedropardomoreno@gmail.com (P.P.M.); 2Interventional Radiology Unit, Hospital Universitario Reina Sofía, 14004 Córdoba, Spain; juanjoseespejo@gmail.com; 3Interventional Radiology Unit, Hospital Universitario y Politécnico La Fe, 46026 Valencia, Spain; raulgamar@gmail.com (R.G.M.); perezenguix.daniel@gmail.com (D.P.E.); martinez.jjo@gmail.com (J.J.M.-R.); 4Management, Monitoring and Data Analysis, Hospital Universitario y Politécnico La Fe, 46026 Valencia, Spain; serena_pisoni@iislafe.es; 5Nuclear Medicine Department, Hospital Universitario Virgen de las Nieves, 18014 Granada, Spain; antonio.rodriguez.fernandez.sspa@juntadeandalucia.es

**Keywords:** transarterial radioembolization, transarterial chemoembolization, colon cancer liver metastasis

## Abstract

**Background**: The present study aims to investigate the superiority of TARE-Y90 in the treatment of liver metastases from colorectal cancer in comparison to DEBIRI and perform a parallel resource consumption study to demonstrate a possible favorable cost-effectiveness balance. **Methods**: The number of subjects included in this study was 46 for TARE-Y90 and 56 in the DEBIRI group. The variables of interest in this study were collected for all selected subjects. Time-to-endpoint outcomes (overall survival, time to progression and time to extra-hepatic progression) were calculated by Kaplan–Meier analysis, reported as medians with 95% confidence intervals and compared between groups by log-rank testing. Values for median time-to-event and 95% confidence intervals were calculated using bootstrapping. **Results**: Categorization into overall response (OR) and no overall response (NOR) revealed a higher percentage of overall responses in the DEBIRI group (52%) compared to TARE-Y90 (24%). The numerical differences observed in certain response categories did not reach statistical significance, indicating a comparable overall response to treatment between the two cohorts based on the m-RECIST criteria. Median overall survival for the TARE-Y90 cohort was 11.3 (95% CI 10.9–18.6) months and 15.8 (95% CI 14.8–22.7) months for the DEBIRI cohort. Log-rank testing showed no statistically significant differences (*p* = 0.53). Median time to hepatic disease progression for the TARE-Y90 cohort was 3.5 (95% CI 3.4–8.1) months and 3.8 (95% CI 3.7–11.1) months for the DEBIRI cohort. Log-rank testing showed no statistically significant differences (*p* = 0.82). An important result of the resource utilization analysis is that TARE-Y90 patients had 1.33 treatments on average per patient, while DEBIRI patients had 3.16 treatments per patient. TARE-Y90 patients also needed fewer days of hospitalization than those in the DEBIRI group. The consequence is that the overall use of resources was higher for DEBIRI in comparison to TARE-Y90. **Conclusions**: Our analysis of the TARE-Y90 and DEBIRI treatments for CRC liver metastases contributes valuable insights into their comparative effectiveness, revealing no significant differences in radiological responses and overall survival. TARE-Y90 showed higher resource utilization, and its potential advantages in patient comfort and average resource consumption per patient warrant consideration.

## 1. Introduction

Liver metastases from colorectal cancer (CRC) pose a substantial clinical challenge in the field of oncology. Globally, CRC ranks as the third most prevalent cancer [1], with 1.8 million new cases reported in 2017. CRC accounts for 10% of all cancer-related deaths, claiming approximately 244,824 lives annually in Europe [2] and 150,000 in the United States [3].

The incidence of CRC is strongly linked to advancing age and higher socio-demographic status [4]. Lifestyle factors and variation in diet, including the consumption of red or processed meats, sedentary behavior, alcohol intake and smoking, play a pivotal role in its development [5,6]. Furthermore, specific individuals carry inherent predispositions to CRC, while identifiable risk factors include conditions such as inflammatory bowel diseases, papillomavirus infection and acquired immunodeficiency syndrome (AIDS). Remarkably, one in four CRC patients receives the diagnosis of stage IV metastatic disease from the outset [7], with a significant majority (70–80%) involving liver metastases [8,9]. Prognosis for CRC patients is profoundly influenced by the disease’s extent at the time of diagnosis, with a striking contrast in five-year survival rates, ranging from 90% for localized cases to a grim 14% for those grappling with liver or other distant metastases.

The liver stands as the primary target for CRC metastasis [10,11], and liver metastases constitute the leading cause of death associated with CRC [12]. A comprehensive and multidisciplinary approach involving surgery, chemotherapy and targeted therapies is vital for enhancing outcomes in patients grappling with this complex condition.

Non-surgical approaches for managing metastases localized to a single organ, most often the liver, offer significant survival benefits beyond systemic therapy alone. When surgical resection is not an option, strategies like image-guided ablation, embolization or stereotactic body radiation therapy (SBRT) become reasonable choices. These decisions should be made in consultation with a multidisciplinary team [13].

One notable non-surgical approach is transarterial radioembolization (TARE), employing Yttrium-90 (Y90) glass microspheres (TheraSphere, Marlborough, MA, USA). This method utilizes brachytherapy to deliver localized beta radiation via tiny glass microspheres containing radioactive Y90. These microspheres are carefully delivered to the liver tumor’s vasculature through hepatic artery catheterization, ensuring precise radiation delivery to the tumor site. TheraSphere effectively penetrates deep into the tumor, delivering therapeutic radiation while minimizing harm to surrounding healthy tissue. Prior TARE and hepatic arteriography was performed to evaluate the arterial anatomy and define hepatic and tumoral vascularization. In the same procedure, 148 MBq (4mCi) of 99mTc-macroaggregated albumin (99mTc-MAA) were injected through selected arteries for the administration of the treatment, obtaining whole-body scintigraphic images and single-photon emission computed tomography (SPECT/CT). Planar scintigraphic images and SPECT/CT were used to calculate the percentage of hepatopulmonary shunt and assess the intra- and extra-hepatic distribution of the particles to detect unidentified extra-hepatic arterial communications on hepatic arteriography, evaluate the perfusion of the hepatic volume to treat (objective), calculate the tumoral/non-tumoral ratio (TNR) and estimate the dose absorbed by the lung. The activity of 90Y-labeled spheres was estimated to achieve a dose absorbed by the liver to treatment of ≥120 Gy. Personalized dosimetry was calculated using the Simplicity 90Y^®^ software (Mirada Medical, Oxford, UK). A hepatopulmonary shunt >20% or a dose absorbed in the lung >30 Gy (50 Gy if the dose was accumulated in consecutive treatments) was considered a relative contraindication for performing the procedure due to the risk of radiation pneumonitis. The focus of treatment was lobar or selective. Patients with unilobar disease underwent a single TARE, and those with bilobar disease underwent two sessions of TARE separated by 2 weeks [14]. This approach offers hope to patients with unresectable, chemorefractory colorectal liver metastases (CRCLMs), potentially delaying disease progression and extending life. It is well tolerated, with mild-to-moderate, transient adverse events, allowing patients to maintain their daily routines [15]. A pivotal phase III trial has demonstrated its potential to delay disease progression when combined with systemic therapy in patients who have advanced beyond first-line treatment [16,17,18]. Another approach involves DEBIRI, whose microspheres enable controlled drug delivery to tumors, reducing systemic side effects and enhancing the consistency of transarterial chemoembolization in the liver. Irinotecan, highly effective for treating colorectal liver metastases, is particularly well suited for transarterial liver therapies due to its efficient extraction by the liver [17]. DEBIRI therapy combines localized chemotherapy with tumor ischemia, precisely delivering cytostatic agents to the arteries supplying the tumor while avoiding dispersion into non-tumor liver tissue and extra-hepatic organs. DEBIRI has demonstrated an acceptable toxicity profile in several prospective studies, with some trials revealing longer overall survival compared to standard treatments. Simultaneous administration of modified FOLFOX and DEBIRI has also shown increased objective responses compared to FOLFOX alone. Chemoembolization is a viable option for select nonresectable patients, especially when systemic chemotherapy has not yielded the desired results. DEBIRI’s selective intra-arterial delivery of irinotecan within tumor arteries, while minimizing drug washout, enables a higher and prolonged intratumoral dose of irinotecan with significantly lower plasma levels. Presently, evidence for DEBIRI is primarily in the salvage setting, but it may hold potential for patients requiring downstaging before surgery [17,18].

## 2. Materials and Methods

The present study aims to investigate the superiority of TARE-Y90 in the treatment of liver metastases from colorectal cancer in comparison to DEBIRI and perform a parallel resource consumption study to demonstrate a possible favorable cost-effectiveness balance. This dual approach seeks to improve health outcomes, specifically in terms of overall survival and time to progression, for patients undergoing these procedures while also conducting a comparative cost analysis to enhance resource allocation efficiency for this patient population.

## 3. Study Design

This is a retrospective, observational and multi-center study based on patient data analysis. A clinical database was created to collect all information of interest (including clinical variables and resource consumption data) from all patients and procedures which met this study’s inclusion criteria. All interventional radiologists involved in performing these procedures had more than 14 years of experience. Data were collected in a dedicated electronic case report form (eCRF) and were dissociated from any patient personal data by means of pseudonymization following Data Management Policies (Table 1).

## 4. Variables

All variables information is presented in the following table (Table 2).

## 5. Health Resource Consumption Analysis

To perform the resource consumption analysis, an average of the utilized hospital materials and resources was computed for each patient in the study. This was calculated by accounting for all resources used during a procedure, which were divided into three distinct categories according to the treatment phases:(1)Pre-procedural resource consumption (which accounts for the preparation of the patient before the intervention);(2)Treatment resource consumption (which includes the treatment itself, surgery room time, staff involved and specific material used in the procedure, including prophylactic drugs administered to the patient);(3)Post-procedure resource consumption (ICU stay and hospitalization length of stay).

We also calculated the number of procedures per patient to account for re-treatments and therefore the possibility of incurring in the same resource usage more than once. All calculations followed the standards of the Spanish Medical Radiology Society (SERAM).

## 6. Statistical Analysis

First, a descriptive statistical analysis was conducted, differentiating between both treatment groups, for data quality control and outlier detection. Descriptive statistics were obtained using Python 3.10 packages “pandas” 1.4.3 and SciPy 1.7.3.

Then, group analyses were carried out to verify the homogeneity of the populations, i.e., lack of previous significant clinical differences between both study groups, using appropriate hypothesis testing tests: the chi-square or Fisher exact test for categorical variables using the R 4.0.5 base package “stats” implementation; Student’s *t*-test or Mann–Whitney *U* for independent continuous variables and paired *t*-test or sign-rank Wilcoxon test for dependent continuous variables using SciPy 1.7.3 implementations.

Time-to-endpoint outcomes (overall survival, time to progression and time to extra-hepatic progression) was calculated by Kaplan–Meier analysis and reported as a median with 95% confidence interval and compared between groups by log-rank testing. Survival analyses were performed using the R package “survival” 3.5–7. Values for median time-to-event and 95% confidence intervals were calculated using bootstrapping. Subjects were censored at the time of loss of follow-up or in the event of undergoing curative treatment (surgical resection or transplant) during follow-up.

## 7. Results

### 7.1. Retrospective Data Search: Subject Inclusion and Exclusion

The health information systems of the hospitals involved in this study were checked for TARE-Y90 and DEBIRI procedures performed during the period 2010–2021 (Figure 1).

Firstly, all subjects found to have undergone TARE-Y90 and DEBIRI procedures were accurately selected for this study following the established inclusion and exclusion criteria. This fundamental initial step allowed us to select 50 subjects treated with TARE-Y90 and 60 subjects treated with DEBIRI, for a total number of 110 treatments included in this study.

In agreement with the research team, a 6 month follow-up was selected, considering the minimum time to verify the subject’s status after the treatment and the result of it. So, loss to follow-up during 6 months after treatment (if the subject was not dead) was verified as the second step of the inclusion/exclusion diagram. A total of two subjects in the TARE-Y90 group and one in the DEBIRI group were excluded. These subjects were probably referred to one of the institutions involved in the project for treatment and were subsequently followed up elsewhere.

Finally, subjects were checked for the exclusion criteria related to “missing data”. Concerning this, most cases involve individuals referred to our hospital for treatment following multiple procedures. Unfortunately, these procedures are not reported in detail, and there is no pre-treatment radiological report specifying the lesion number and diameter. Additionally, these cases lack comprehensive health records that would include essential baseline covariates.

Overall, the number of subjects included in this study was 46 for TARE-Y90 and 56 in the DEBIRI group. The variables of interest in this study were collected for all selected subjects (46 in the TARE-Y90 group and 56 in the DEBIRI group).

Moreover, we noted that the TARE-Y90 group was significantly more likely to have undergone PET-CT pre-treatment than the DEBIRI group. Nevertheless, these data were lacking for 51% of the subjects, so it was not possible to use the SUV-related variables to assess their prognostic and descriptive effectiveness for the characteristics of the study subjects. However, all other variables included in the calculation of these risk stratification scores are detailed in the study population description tables and used for descriptive statistical analysis.

Summary statistics for the study population are shown in Table 3. No statistically significant differences could be observed between patients treated with TARE-Y90 and with DEBIRI, either in socio-demographic and clinical variables or in variables related to the imaging study.

Of note, carcinoembryonic antigen (CEA) value data registered in the pre-treatment blood analysis showed a very large range and high variability among patients in both treatment groups. Therefore, even though this variable did not reach statistical significance, the results should be interpreted with caution due to the substantial individual variations observed.

The DEBIRI group was found to have undergone significantly more primary tumor surgeries than the TARE-Y90 group. However, these data were not considered relevant or prognostic for a relevant difference between the two groups in terms of the previous treatment the patients had undergone for the primary tumor, as, in both groups, the use of systemic therapy was found in almost all participants, and at least one of the two types of previous treatment had recurred in 100% of the participants.

After conducting the analysis of descriptive statistics and standardized mean differences, the groups were considered reasonably comparable, with no evidence of substantial covariate imbalance between the TARE-Y90 and DEBIRI treatment groups and without the need for additional adjustment. This result confirmed the accurate selection of subjects suitable for this study, performed following the established inclusion and exclusion criteria. As a drawback, the total number of treatments included in this study was small. Small sample sizes may lead to imprecise estimates when additional adjustment is performed, limiting their reliability and reducing the statistical power of the analysis.

### 7.2. Response to Treatment

The primary objective of this study was to compare radiological responses to the TARE-Y90 and DEBIRI treatments between the two cohorts. Radiological response was assessed according to the mRECIST criteria, as indicated for loco-regional treatment of colorectal liver metastasis by the EASL. Given that the timing of response assessment is not standardized by international guidelines and was different in the institutions involved in this study after TARE-Y90 and DEBIRI treatments, analyses were performed according to Best Overall Response criteria.

Radiological response rates were reported and analyzed in the four classes of mRECIST—complete response (CR), partial response (PR), stable disease (SD) and progressive disease (PD)—and were also classified into overall response (OR), including CR and PR, or no overall response (NOR), including SD and PD. The results are shown in Table 4 and Table 5.

Statistical analyses using Fisher’s exact test did not reveal significant differences in the rate of response to treatment between the TARE-Y90 and DEBIRI groups. Examining the response categories individually, we observed numerical variations between the TARE-Y90 and DEBIRI groups. Specifically, the DEBIRI group demonstrated a higher percentage of complete responses (CRs), at 12%, compared to TARE-Y90, at 4%. However, it is important to note that this difference did not reach statistical significance (*p* = 0.06). The percentages of partial response (PR), stable disease (SD) and progressive disease (PD) showed no substantial differences between the two treatment cohorts. The categorization into overall response (OR) and no overall response (NOR) revealed a higher percentage of overall responses in the DEBIRI group (52%) compared to TARE-Y90 (24%). Yet, like the CR category, this difference did not achieve statistical significance (*p* = 0.19).

In summary, this study did not find significant differences in radiological response rates between the TARE-Y90 and DEBIRI groups. The numerical differences observed in certain response categories did not reach statistical significance, indicating a comparable overall response to treatment between the two cohorts based on the m-RECIST criteria.

### 7.3. Time-to-Event Analyses

In addition to evaluating the Best Overall Response, the second objective of this study was to compare time-to-event analysis results, including overall survival (OS), hepatic progression-free survival (hPFS), progression-free survival (PFS) and tumor response, between TARE-Y90 and DEBIRI in patients with liver metastases from CRC.

OS was calculated from the initiation of the primary treatment until the date of death or the last follow-up. hPFS and PFS were determined retrospectively, starting from the initial regional treatment until the first documented evidence of liver progression (hPFS) or progression anywhere in the body (PFS). Global PFS included disease progression at any anatomical site, such as the brain, liver, lung and lymph nodes, not limited to the liver.

**(a)** 
**Overall survival (OS)**


The Kaplan–Meier curve for overall survival is shown in Figure 2. OSorresponds to the time from the initial regional treatment until the date of death or the last follow-up.

Median overall survival was 11.3 (95% CI 10.9–18.6) months for the TARE-Y90 cohort and 15.8 (95% CI 14.8–22.7) months for the DEBIRI cohort. Log-rank testing showed no statistically significant differences (*p* = 0.53).

In conclusion, no significant differences in terms of overall survival (the primary endpoint in this retrospective observational study) were found between the two cohorts in our study.

**(b)** 
**Hepatic progression-free survival (hPFS)**


The Kaplan–Meier curve for time to hepatic disease progression (by mRECIST criteria) is shown in Figure 3. hPFS corresponds to the time from the initial regional treatment until the first documented evidence of liver progression.

Median time to hepatic disease progression was 3.5 (95% CI 3.4–8.1) months for the TARE-Y90 cohort and 3.8 (95% CI 3.7–11.1) months for the DEBIRI cohort. Log-rank testing showed no statistically significant differences (*p* = 0.82).

In conclusion, no significant differences in terms of time to hepatic disease progression were found between the two cohorts in our study.

**(c)** 
**Progression-free survival (PFS)**


The Kaplan–Meier curve for time to disease progression (by mRECIST criteria) is shown in Figure 4. PFS corresponds to the time from the initial regional treatment until the first documented evidence of progression anywhere in the body.

Median time to progression was 3.4 (95% CI 3.3–7.7) months for the TARE-Y90 cohort and 3.8 (95% CI 3.6–9) months for the DEBIRI cohort. Log-rank testing showed no statistically significant differences (*p* = 0.86).

As demonstrated in this section, the populations exhibit comparability across all time-to-event endpoint analyses, as no significant differences were demonstrated by log-rank testing. In essence, both treatment cohorts display similar outcomes across a range of time-to-event endpoints.

### 7.4. Health Resource Consumption Analysis

This part of the analysis accounted for all resources used in the interventions: pre-procedure (Table 6), procedure (Table 7) and post-procedure (Table 8). Finally, we also include an analysis of the re-treatments needed by patients (Table 9).

As seen in Table 6, the results indicate that the TARE-Y90 treatment requires a pre-procedural phase, which DEBIRI does not. However, this phase is critical for the TARE-Y90 treatment to be effective and ensures a significantly greater technical outcome of the procedure.

This can be seen in the post-procedure phase (Table 8), in which it is observed that patients undergoing DEBIRI require more than 2 days in the intensive care unit on average, while it is not necessary for TARE-Y90. In addition, when analyzing the length of stay, TARE-Y90 patients also need fewer days of hospitalization than patients undergoing DEBIRI.

An important result of the resource utilization analysis is that TARE-Y90 patients have, on average, 1.33 treatments per patient, while DEBIRI patients have 3.16 treatments per patient (Table 9). This is a significant difference in favor of TARE-Y90, since DEBIRI patients need to undergo treatment nearly three times more often. The consequence is that the overall use of resources is higher for DEBIRI in comparison to TARE-Y90. This affects the organization of hospitals (number of available beds, waiting lists) and patients’ quality of life.

## 8. Discussion

Our study aimed to compare health outcomes and resource consumption between cohorts of TARE-Y90 and DEBIRI patients with liver metastases from colorectal cancer (CRC). We employed a robust retrospective data search methodology within the framework of a multicentric project.

The initial steps involved a meticulous descriptive statistics analysis and standardized mean differences assessment, crucial for establishing the comparability of the treatment groups. The results reassuringly indicated reasonable comparability, affirming the precise selection of subjects based on predefined inclusion and exclusion criteria.

No statistically significant differences could be observed between patients treated with TARE-Y90 and with DEBIRI, either in terms of socio-demographic and clinical variables or in terms of variables related to the imaging study. Of note, carcinoembryonic antigen (CEA) value data registered in the blood analysis pre-treatment showed a very large range and high variability among patients in both treatment groups. Therefore, even though this variable did not reach statistical significance, the results should be interpreted with caution due to the substantial individual variations observed (Table 3). In summary, this study did not find significant differences in the radiological response rates between the TARE-Y90 and DEBIRI groups. The numerical differences observed in certain response categories did not reach statistical significance, indicating a comparable overall response to treatment between the two cohorts based on mRECIST criteria.

The primary objective of this study was to investigate the radiological response rate, categorized according to mRECIST criteria, and to compare the effectiveness of the TARE-Y90 and DEBIRI treatments. However, the results do not demonstrate significant differences. Notably, while there was a numerical difference in the CR category favoring DEBIRI, this did not reach statistical significance (*p* = 0.06). Similarly, the overall response rates did not show a significant difference (*p* = 0.19), despite a higher percentage in the DEBIRI group. Therefore, despite variations in specific response categories, the global treatment responses were comparable.

The numerical differences observed may prompt further exploration, especially given the potential implications for patient outcomes. However, the lack of statistical significance highlights the need for cautious interpretation, and there is a possibility that these differences may be due to chance or other factors not captured in our analysis, such as genetic mutations (KRAS).

Another important outcome was the comparative analysis of resource consumption between TARE-Y90 and DEBIRI in patients with liver metastases from CRC. Our findings indicate that DEBIRI necessitates more procedures per patient and therefore entails a higher total health resource consumption.

Consequently, we assert that TARE-Y90 may offer advantages over DEBIRI for both the patient and the healthcare facility. For the patient, this advantage is evident in terms of reduced discomfort due to a lower number of treatments, and for the hospital, in consideration of the average total consumption of healthcare resources per patient. Radiation dosimetry has evolved to become more tailored to the patient and the target lesion (a trend in recently included patients), with treatment dose and distribution adapted for palliation, bridging or downstaging to liver transplantation, converting to surgical resection candidacy, etc. Recently published data thus far reinforce the conclusion that “personalizing” dosimetry yields real-world improvements in tumor response and overall survival while maintaining a favorable adverse event profile [19,20,21,22].

Re-treatments have a significant impact on patients’ quality of life (QoL), as patients were observed to undergo several additional lines of chemoembolization with DEBIRI, as well as having an impact in terms of hospital resource consumption metrics. Given that DEBIRI patients come back for re-treatments, it may also impact waiting lists.

In studies regarding economic evaluations of TARE-Y90, it was noted that the inclusion of Y-90 TARE therapy was associated with additional cost [23,24,25,26,27,28,29,30], mainly because it was compared with well-known and low-cost chemotherapy drugs such as HAI (hepatic artery infusion) [31,32] and FOLFOX (folinic acid, fluorouracil and oxaliplatin) [33] or BSC (“best supportive care”) [23,24,25,26], therapies aimed at treating symptoms.

Despite additional cost (ranging from USD 16,824 [29] to 25,320 [24] PPP), Y-90 TARE therapy has demonstrated advantages in improving efficiency, such as reducing hospital stay (2 days for Y-90 TARE vs. 9 days for HAI) [29]; improving health outcomes (Y-90 TARE vs. BSC [23,24,25,26] or HAI [27]); improving “years of life gained” (LYG) (Y-90 TARE versus BSC, 1.12 [19,21] to 1.35 [24], and Y-90 TARE versus HAI, 0.37); and quality-adjusted life years (QALY) (Y-90 TARE versus BSC, 0.81 [23,25,26] to 0.83 [24], and Y-90 TARE + HAI versus HAI, 0.35 [27]). Although a retrospective study of Y-90 TARE versus HAI [29] showed longer overall survival for HAI (16.3 vs. 31.2 months), the study reported a lower probability of survival, as there were more patients in the TARE group with previous liver resection at the time of diagnosis.

Fusco et al. [27] evaluated the use of Y-90 TARE in first-line treatment for chemotherapy-naïve patients and identified restricted information on primary care resource costs as a limitation. The remaining economic studies [23,24,25,26,27,29] evaluated the use of Y-90 TARE in successive lines of treatment for chemotherapy-refractory patients, drawing clinical data from two retrospective studies [29,30] and one clinical trial [31]. The first retrospective study by Bester et al. [30] had a representative population (*n*: 339) for Y-90 TARE and was used in four [23,24,25,26] full economic evaluations. The second retrospective study, by Dhir et al. [29], was used to estimate the cost of HAI and Y-90 TARE treatment in the same reference and evaluated a smaller population (*n*: 49). Furthermore, the clinical trial by Gray et al. [31] evaluated a population of 35 patients and was used to compare Y-90 TARE + HAI versus HAI [27].

Given the indications for treatment with Y-90 TARE by the American Society of Interventional Radiology [33], which focus on including patients with surgically unresectable liver and hepatic neoplasms, appropriate patient selection is relevant for optimal outcome. As such, the combination of Y-90 TARE with systemic chemotherapy treatment is not recommended as a first-line treatment in patients with unresectable mCRC [32]. However, the addition of Y-90 TARE to standard second-line chemotherapy (as demonstrated in the phase III EPOCH clinical trial) [16] has demonstrated improved progression-free survival (PFS) and liver disease-free survival (hepatic PFS), further supporting the cost-effectiveness advantage of Y-90 TARE therapy in patients with unresectable mCRC.

Studies reported costs in different currencies and reference years [24,25,29], limiting the comparability of the results. In our study, these were converted to 2020 (USD PPP cost) to address this issue. Although this is a global review system, most economic evaluations were conducted from a European perspective, which may limit the external validity of our review for other countries.

Moving to the second objective of this study, our meticulous examination of the time-to-event analyses provides valuable insights into the comparative effectiveness of TARE-Y90 and DEBIRI in patients with CRC liver metastases. The observed comparability between TARE-Y90 and DEBIRI in overall survival, hepatic progression-free survival and progression-free survival implies that the choice between these treatments may not significantly impact these outcomes in CRC patients with liver metastases.

Despite this study’s methodological rigor, it is essential to acknowledge a limitation due to the relatively small sample size of the included treatments. The inherent challenges associated with small sample sizes, particularly in terms of precision and statistical power when making additional adjustments, are recognized and considered in the interpretation of our findings.

In conclusion, our study comparing TARE-Y90 and DEBIRI treatments for CRC liver metastases contributes valuable insights into their comparative effectiveness, revealing no significant differences in radiological responses and overall survival. Despite numerical variations, the overall treatment responses were comparable, emphasizing caution in interpretation due to sample size limitations. While TARE-Y90 showed higher resource utilization, its potential advantages in patient comfort and average resource consumption per patient warrant consideration. Our findings offer insights for clinical decision-making while acknowledging this study’s retrospective nature and the need for prospective validation. Our study opens avenues for future research that can build on these results, emphasizing the importance of continued research and potentially larger cohorts to elucidate the nuanced aspects of treatment responses in patients with liver metastases from CRC.

## Figures and Tables

**Figure 1 diagnostics-15-00796-f001:**
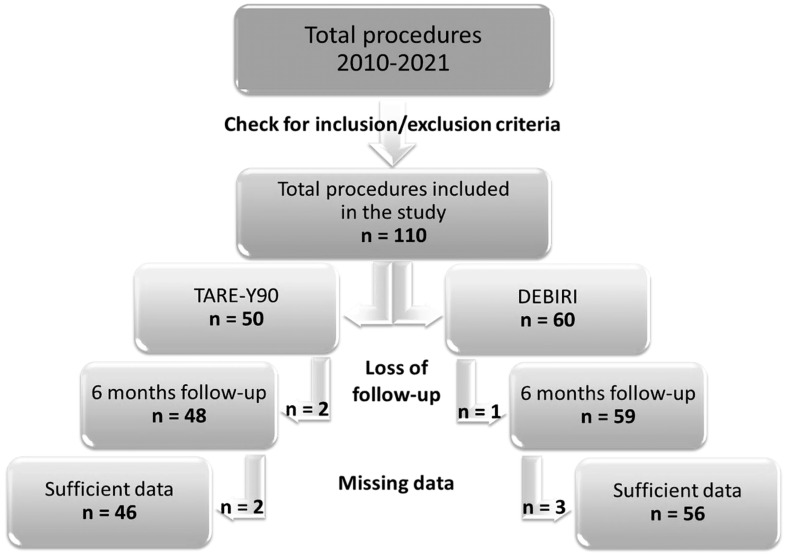
Flow diagram of this study.

**Figure 2 diagnostics-15-00796-f002:**
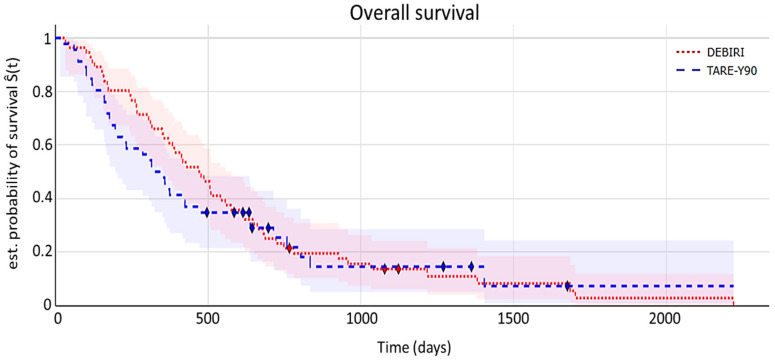
Kaplan–Meier curve illustrating overall survival. The shaded areas depict 95% confidence intervals. The symbol ⧫ denotes censored data.

**Figure 3 diagnostics-15-00796-f003:**
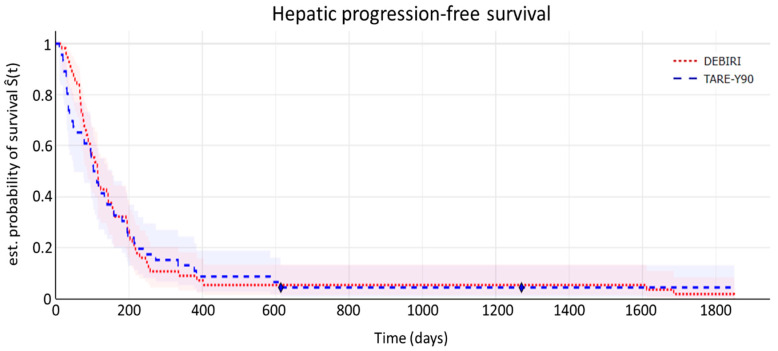
Kaplan–Meier curve illustrating hepatic progression-free survival time. The shaded areas depict 95% confidence intervals. The symbol ⧫ denotes censored data.

**Figure 4 diagnostics-15-00796-f004:**
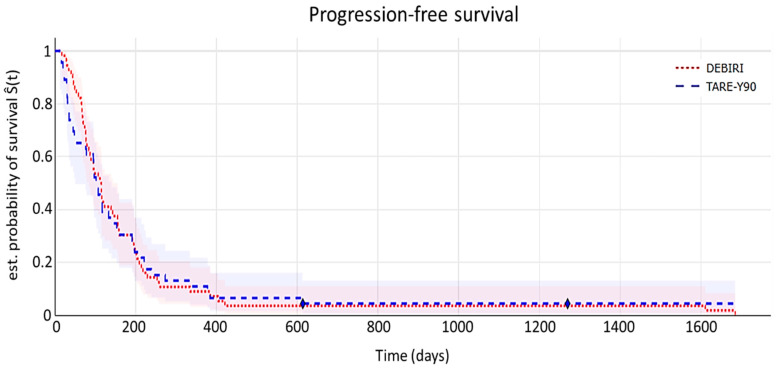
Kaplan–Meier curve illustrating progression-free survival time. The shaded areas depict 95% confidence intervals. The symbol ⧫ denotes censored data.

**Table 1 diagnostics-15-00796-t001:** Inclusion and exclusion criteria.

Inclusion Criteria	Exclusion Criteria
Colorectal cancer with unilobar or bilobar metastatic liver disease (stage IV), unresectable and with disease progression in the liver after several lines of chemotherapy (at least one).	Background of hepatic encephalopathy.
Clinically stable or resected primary tumor.	Pulmonary insufficiency or clinically evident chronic obstructive pulmonary disease.
Tumor replacement below 50% of the total volume of the liver.	Cirrhosis and portal hypertension.
Patients treated with TARE-Y90 (TheraSphere^®^ Y-90 Glass Microspheres- Boston Scientific, Malborough, MA, USA).	History of severe allergy or intolerance to contrast agents, narcotics, sedatives or atropine that cannot be treated medically.
Patients treated with DEBIRI (DC Beads M1 70–150 µ- Boston Scientific, Malborough, MA, USA).	Contraindications to angiography or selective visceral catheterization such as bleeding or coagulopathy not controllable with common hemostatic agents (e.g., device closure).
Life expectancy greater than 6 months at the start of locoregional therapy (minimum post-procedure follow-up time: 6 months or until death).	Intervention or compromise of the Ampulla of Vater.
ECOG 0–1 until the first treatment under scope.	Clinically obvious ascites (traces of ascites on imaging are acceptable).
Creatinine serum < 2.0 mg/dL.	Hepatic toxicity due to previous cancer therapy that has not been resolved before the start of study treatment, if the researcher determines that the continuing complication will compromise the patient’s safe treatment.
Serum bilirubin up to 1.2× upper limit.	Significant life-threatening extra-hepatic disease, including patients who are on dialysis, have unresolved diarrhea or have severe unresolved infections, including patients known to be HIV-positive or to have acute HBV or HCV.
Albumin > 3.0 g/dL.	Confirmed extra-hepatic metastases. Limited and indeterminate extra-hepatic lesions are allowed in the lung and/or lymph nodes (up to 5 lesions in the lung, with each individual lesion < 1 cm; any number of lymph nodes with each individual node < 1.5 cm).
Neutrophil count > 1200/mm^3^ (1.2 × 10^9^/L)	Previous treatment with liver radiotherapy.
	Previous intra-arterial therapy directed at the liver, including transarterial chemoembolization (TACE) using irinotecan-loaded beads or TARE-Y90 therapy.
	Treatment with biological agents within 28 days of receiving TARE-Y90 therapy.

**Table 2 diagnostics-15-00796-t002:** Study variables.

▪ Socio-demographic variables: • Age • Sex	▪ Clinical variables: • Comorbidities: high blood pressure (HBP), diabetes mellitus (DM), dyslipidemia (DL) • Carcinoembryonic antigen (CEA) • Tumor size/volume • Extra hepatic involvement • Previous treatments (surgery, systemic therapy)
▪ Response variables: • Best Overall Response by image according to RECIST 1.1 • Tumor response according to RECIST 1.1 criteria every 1–2 months up to a minimum of 6 months of follow-up or death • Progression-free survival up to a minimum of 6 months of follow-up or death • Hepatic progression-free survival up to a minimum of 6 months of follow-up or death • Overall survival up to a minimum of 6 months of follow-up or death • Major and minor complications up to 90 days after the procedure	▪ Variables related to resource consumption: • Previous chemotherapy treatments • Number of procedures (number of TARE-Y90/DEBIRI treatments) per patient • Performance and material related to the procedure per patient • Treatment dose (TARE Y90: target site dose in Gy; DEBIRI: mg irinotecan) • Hospital stays per patient (days) • Length of stay in ICU per patient (hours), if applicable • Complications per patient • Follow-up visits

**Table 3 diagnostics-15-00796-t003:** Description of study population. Numerical variables are shown as mean ± standard deviation (range). Categorical variables are shown as relative frequency (absolute frequency). *p*-value calculation: a = Student’s *t*-test; b = Mann–Whitney’s *U* test; c = Fisher’s exact test. Statistical significance *, **. HTA: arterial hypertension; CEA: carcinoembryonic antigen; CT: computed tomography; DL: dyslipidemia; DM: diabetes mellitus; ECOG-PS: Eastern Cooperative Oncology Group, PS: Performance Status; FOLFIRI:folinic acid, fluorouracil and irinotecan hydrochloride; FOLFOX: folinic acid, fluorouracil and oxaliplatin; LM: liver metastasis; MRI: magnetic resonance imaging; PET-CT: positron emission tomography; PT: primary tumor; TACE: transarterial chemoembolization.

		TARE-Y90 (*N* = 46)	DEBIRI (*N* = 56)	*p*
Age (years)		62 ± 10 (34–77)	62 ± 9 (45–79)	0.89 (a)
Sex	Men	67% (31)	70% (39)	0.83 (c)
	Women	33% (15)	30% (17)	
ECOG	PS 0	72% (33)	66% (38)	0.56 (c)
	PS 1	26% (12)	25% (14)	
	PS 2	2% (1)	7% (4)	
HTA		46% (21)	41% (23)	0.69 (c)
DM		13% (6)	16% (9)	0.78 (c)
DL		22% (10)	30% (17)	0.37 (c)
PT localization	DX	28% (13)	23% (13)	0.81 (c)
	SX	72% (33)	75% (42)	
	DX; SX	0% (0)	2% (1)	
PT vascular invasion		50% (23)	28% (16)	0.30 (c)
RAS mutation		28% (13)	27% (15)	1.00 (c)
BRAf		2% (1)	4% (2)	1.00 (c)
LM at PT diagnosis		72% (33)	70% (39)	0.83 (c)
Other Metastasis during the treatment	Lymph node	12% (6)	12% (7)	1.00 (c)
	Lung	11% (5)	12% (7)	
	Other location	5% (2)	4% (2)	
	No	72% (33)	72% (40)	
Ascites		4% (2)	0% (0)	0.20 (c)
Peritoneal carcinomatosis		7% (3)	2% (1)	0.32 (c)
Imaging test mode	CT	80% (37)	84% (47)	0.78 (c)
	PET-CT	9% (4)	5% (3)	
	RM	7% (3)	9% (5)	
	Echography	4% (2)	2% (1)	
N° LM		4 ± 2	4 ± 4	0.61 (b)
Size of target lesion		45.7 ± 33.8	53.1 ± 35.1	0.30 (a)
PET-CT pre-treatment		61% (28)	39% (22)	0.05 (c) *
LM localization	LHD	50% (23)	34% (19)	0.14 (c)
	LHI	2% (1)	9% (5)	
	Bilobar extension	48% (22)	56% (32)	
Tumor burden	<25%	58% (27)	68% (38)	0.58 (c)
	from 25 to 50%	33% (15)	23% (13)	
	>50%	9% (4)	9% (5)	
Pre-treatment CEA value (ng/mL)		286 ± 958	179 ± 557	0.61 (a)
Previous treatment for PT	Primary tumor surgery	78% (36)	96% (54)	0.006 (c) **
	Systemic therapy	96% (44)	100% (56)	0.20 (c)
	Surgery and/or systemic therapy	100% (46)	100% (56)	1.00 (c)
N° lines of systemic therapy		1.9 ± 0.7	2.1 ± 1.4	0.57 (a)
Type of first-line therapy	FOLFOX	65% (30)	66% (37)	0.92 (c)
	FOLFIRI	15% (7)	18% (10)	
	FOLFOX; FOLFIRI	20% (9)	16% (9)	
Type of previous treatment for LM	Surgery	30% (14)	50% (28)	0.07 (c)
	TACE	2% (1)	9% (5)	1.00 (c)
	Ablation	20% (9)	21% (12)	1.00 (c)

**Table 4 diagnostics-15-00796-t004:** Response to treatment in both treatment cohorts, classified into CR/PR/SD/PD according to mRECIST. Statistical differences were evaluated with Fisher’s exact test. CR = complete response; PR = partial response; SD = stable disease; PD = progressive disease.

	TARE-Y90 (*n* = 46)	DEBIRI (*n* = 56)	*p*
CR	4% (1)	12% (9)	0.06
PR	20% (9)	20% (10)	
SD	30% (15)	20% (10)	
PD	46% (21)	48% (27)	

**Table 5 diagnostics-15-00796-t005:** Response to treatment in both treatment cohorts, classified into OR/NOR according to mRECIST. Statistical differences were evaluated with Fisher’s exact test. OR = overall response; NOR = no overall response.

	TARE-Y90 (*n* = 46)	DEBIRI (*n* = 56)	*p*
OR	24% (10)	52% (19)	0.19
NOR	18% (36)	48% (37)	

**Table 6 diagnostics-15-00796-t006:** Average pre-procedure resource consumption.

Device/Procedure/Health Provider/Time	Units per TARE-Y90 Procedure	Units per DEBIRI Procedure
Selective angiographic catheter	1	0
Simmons catheter	1	0
Guide	1	0
Introducer	1	0
Vascular puncture needle	1	0
Micro catheter	1	0
Vascular closure device	1	0
Coils	1	0
99Tc-MAA	1	0
Pre-treatment assessment with MAA	1	0
Dosimetry of patients treated with radioactive isotopes	1	0
Physician/specialist	3	0
Specialist technician	1	0
Nurse	2	0
Hospitalization (days)	1	0

**Table 7 diagnostics-15-00796-t007:** Average procedure resource consumption.

Device/Drug/Health Provider/Time	Units per TARE-Y90 Procedure	Units per DEBIRI Procedure
Selective angiographic catheter	1	1
Simmons catheter	1	1
Guide	1	1
Introducer	1	1
Vascular puncture needle	1	1
Micro catheter	1	1
Vascular closure device	1	1
90Y particles	1	0
DEBIRI particles	0	2
Irinotecan	0	2
- Gastroduodenal ulcer prophylaxis	1	1
- Anti-nausea prophylaxis	0	1
- Postembolization syndrome prophylaxis	1	1
- Intravenous corticosteroid before treatment	1	1
- Cefuroxime	1	1
Physician/specialist	3	1
Specialist technician	1	1
Nurse	2	2

**Table 8 diagnostics-15-00796-t008:** Average post-procedure resource consumption.

Procedure/Hospitalization	Units per TARE-Y90 Procedure	Units per DEBIRI Procedure
Whole-body positron tomography (PET-CT)	1	0
Dosimetry of patients treated with radioactive isotopes	1	0
Stay in intensive care unit (hours)	0	2.20
Hospitalization (days)	1	2.56

**Table 9 diagnostics-15-00796-t009:** Average treatments per patient.

Concept	TARE-Y90	DEBIRI
Average treatments per patient	1.33	3.16

## Data Availability

The original contributions presented in this study are included in the article. Further inquiries can be directed to the corresponding author.

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
