# Peer review of "Health Outcomes and Resource Consumption Analysis of Radioembolization with Y90 Glass Microspheres (TARE-Y90) Versus Transarterial Chemoembolization with Irinotecan (DEBIRI) in Patients with Liver Metastases from Colorectal Cancer in Spain"

_diagnostics, 2025, doi:10.3390/diagnostics15070796_

Round 1
Reviewer 1 Report
Comments and Suggestions for Authors
In the manuscript „Health Outcomes and Resource Consumption …“ the authors Ciampi JJ et al compare DEB-TACE with 90Y glass TARE for the treatment of liver metastases of colorectal cancer in spain. They retrospectively compared outcome after treatment and utilized resources in a multi-center analysis in Spain.
They were able to show that both therapies showed a relatively comparable outcome with respect to tumor control (complete and partial response and stable disease) and hepatic progression free survival. There was a trend towards longer overall survival for the TACE group. Resources used for TARE were higher than for TACE, however, a higher number of TACE procedures per patient was observed in the clinic counterbalancing the higher resource utilization from TARE. Additionally, patients stayed for a much shorter time on the ward and did not need intensive care with TARE.
The manuscript is well written, shows a lot of very interesting data when comparing the two therapies and is overall a welcome addition to the evidence pertaining to directed liver cancer therapies.
I would like to point out several areas where the manuscript would benefit (in my opinion) from changes/improvements:
- TARE is not a therapy where there is *one* standard procedure to follow but where there is a significant variation in approach depending on country/hospital. TARE is also rapidly evolving over time; the use of dosimetry for treatment planning comes to mind. The authors may give some more information on the details of the TARE procedure: (1) how was injected activity determined (single-compartment model, partition model or voxel-wise dosimetry?, intended dose for tumor? Used methodology for dosimetry?). Who injects the spheres (#of physicians (3) in Table 6 seems high)? Post-therapy imaging of sphere distribution (PET-CT or bremsstrahl scintigraphy?), What is meant with whole-body PET-CT (TARE-PET or other?, time for data acquisition considering the low 90Y statistics).
- In a retrospective study a referral bias cannot be excluded. The authors have gone a long way in proving that both groups are comparable. They may also comment on the criteria for sending patients to TARE or TACE in the interdisciplinary tumor board.
- Authors may comment on side-effects apart from longer time on the ward and need for intensive care for TACE patients.
- The author often equate „no statistically significant difference“ and „comparable“. I would suggest a more cautious interpretation, e.g. overall survival showed a trend for longer survival of post-TACE patients.
- Post-therapeutic tumor dosimetry data would be a very interesting for these patients in order to interpret TARE results. It is well known that achieved dose is related to outcome and that the TARE field goes to higher doses if possible. The image data might be there to do dosimetry retrospectively.
Some point-by-point comments:
Page 1, Abstract:
„Survival analysis were performed using R package „survival“. I would remove this from the abstract.
Page 2, Abstract:
There seems to be a contradiction or a difficult to understand sentence: Use of resources for DEBIRI is higher but TARE showed higher resource utilization? What does that mean. Please rephrase.
Page 3, 3rd paragraph:
Does the study really aim to investigate the superiority of TARE or simply to compare the two?
Page 6:
List of variables and table on next page are redundant.
Page 10, Table 3:
Several abbreviations are not in the abbreviation list. Several abbreviations may be written out to improve reading. PET-TC à PET-CT. Tumor burden: <25%, from 25 to 50%, >50 %?
Pages 14-16, Table 6/7/8:
See above. Depends on local procedure that may well be different at different hospital sites, countries etc.
Author Response
1. TARE is not a therapy where there is *one* standard procedure to follow but where there is a significant variation in approach depending on country/hospital. TARE is also rapidly evolving over time; the use of dosimetry for treatment planning comes to mind. The authors may give some more information on the details of the TARE procedure: (1) how was injected activity determined (single-compartment model, partition model or voxel-wise dosimetry?, intended dose for tumor? Used methodology for dosimetry?). Who injects the spheres (#of physicians (3) in Table 6 seems high)? Post-therapy imaging of sphere distribution (PET-CT or bremsstrahl scintigraphy?), What is meant with whole-body PET-CT (TARE-PET or other?, time for data acquisition considering the low 90Y statistics).
Patients were evaluated by a multidisciplinary team where were proposed as candidates for TARE. First, 2-3 weeks prior TARE and before evaluation of the arterial tree in the CT study, hepatic arteriography was performed to evaluate the arterial anatomy and define hapatic and tumoral vascularization. Arterial vessels that could condition the Y90 spheres reaching extrahepatic organs (fundamentally the gallbladder and the antropiloric gastric regions) were embolized. Thereafter, in the same procedure, 148 MBq (4mCi) of 99mTc-macroaggregated albumin (99mTc-MAA) were injected through selected arteries for the administration of the treatment, obtain-ing whole body scintigraphic images and single photon emission computed tomography (SPECT/CT) of the abdomen before 2 h after the injection to avoid the appearance of free technetium.
Planar scintigraphic images and SPECT/CT were used to calculate the percentage of hepatopulmonary shunt and asssess the intra- and extrahepatic distribution of the particles to detect unidentified extrahepatic arterial communications in the hepatic arteriography, evaluate the perfusion of the hepatic volume to treat (objective), calculate the tumoral/non tumoral ratio (TNR) and estimate the dose absorbed by the lung. The activity of90Y labeled spheres was estimated to achieve a dose absorbed bythe liver to treatment ≥120 Gy.
Personalized dosimetry was calculated using the Simplicity® software (Mirada Medical). A hepatopulmonary shunt >20% or a dose absorbed in the lung >30 Gy (50 Gy if the dose was accumualted in consecutive treatments) was considered as a relative contraindication for performing the procedure due to the risk of radiation pneumonitis. The focus of treatment was lobar or selective. Patients with unilobar disease underwent a single TARE, and those with bilobar disease underwent two sessions of TARE separated by 2 weeks. Finally, a90Y PET/CT study was performed in all the cases in order to visualize the distribution of the 90Y labeled spheres in the hepatic volume to treat and rule out the presence of extrahepatic foci.
2. In a retrospective study a referral bias cannot be excluded. The authors have gone a long way in proving that both groups are comparable. They may also comment on the criteria for sending patients to TARE or TACE in the interdisciplinary tumor board.
This situation was in relation with each hospital availability. Remember this is a retrospective study different moments and different centers.
3. Authors may comment on side-effects apart from longer time on the ward and need for intensive care for TACE patients.
In a several cases (4), these patients needed UCI support to pain management, related to idiosyncrasy of some hospitals in Spain.
4. The author often equate „no statistically significant difference“ and „comparable“. I would suggest a more cautious interpretation, e.g. overall survival showed a trend for longer survival of post-TACE patients.
Ok. Agree.
5. Post-therapeutic tumor dosimetry data would be a very interesting for these patients in order to interpret TARE results. It is well known that achieved dose is related to outcome and that the TARE field goes to higher doses if possible. The image data might be there to do dosimetry retrospectively.
Agree
Some point-by-point comments:
Page 1, Abstract:
„Survival analysis were performed using R package „survival“. I would remove this from the abstract.
DONE
Page 2, Abstract:
There seems to be a contradiction or a difficult to understand sentence: Use of resources for DEBIRI is higher but TARE showed higher resource utilization? What does that mean. Please rephrase.
DONE
Page 3, 3rd paragraph:
Does the study really aim to investigate the superiority of TARE or simply to compare the two?
The study aim to investigate the superiority of TARE.
Page 6:
List of variables and table on next page are redundant.
Ok. Removed.
Page 10, Table 3:
Several abbreviations are not in the abbreviation list. Several abbreviations may be written out to improve reading. PET-TC à PET-CT. Tumor burden: <25%, from 25 to 50%, >50 %?
Ok. Done.
Pages 14-16, Table 6/7/8:
See above. Depends on local procedure that may well be different at different hospital sites, countries etc.
These findings reflect a ´real life´ in some of the Spanish representative hospitals that participate in this study.
Reviewer 2 Report
Comments and Suggestions for Authors
- How does higher socio-demographic status increase the incidence of CRC?
- Can authors be specific with the alcohol intake. How much is likely to increase the risk of CRC?
- How precise is TheraSphere and is there any reported limitations with this approach?
- If DEBIRI has the potential to reduce systematic side effects, does it mean it is better than TheraSphere?
- How were TARE-Y90 and DEBIRI produced?
- Which one will authors go for based on your study, TARE-Y90 or DEBIRI?
- Why did authors use only age and sex for their socio-economic variables?
- Was patient's tumor size measured?
Author Response
Comments and Suggestions for Authors
-Revisor 2
1. How does higher socio-demographic status increase the incidence of CRC?
This information is not the propose in this study.
2. Can authors be specific with the alcohol intake. How much is likely to increase the risk of CRC?
This information was not analyzed in this study. Study is related to colorectal cancer liver metastasis, not hepatocarcinoma.
3. How precise is TheraSphere and is there any reported limitations with this approach?
Ok. Done. The detailed technique explanation was included in text.
4. If DEBIRI has the potential to reduce systematic side effects, does it mean it is better than TheraSphere?
Both are techniques indicated in colorectal cancer metastasis after systemic chemotherapy lines and they have specific side effects. Study cannot demonstrate superiority of DEBIRI over TARE or viceversa.
5. How were TARE-Y90 and DEBIRI produced?
Sorry, I didn´t understand this question.
6. Which one will authors go for based on your study, TARE-Y90 or DEBIRI?
One hospital was dedicated to TARE
One hospital was dedicated to DEBIRI
One hospital was dedicated to DEBIRI and TARE
7. Why did authors use only age and sex for their socio-economic variables?
Yes, These variables were included in the study.
8. Was patient's tumor size measured?
Yes, Tumoral size mean were 4-5 cm in dominant lesion.
Reviewer 3 Report
Comments and Suggestions for Authors
- Table 3. There are many space to show whole typing of abbreviations over the left colum, or the abbreviations should be presented below the table, it is difficult to get a good idea at a glance.
- Is PET-CT pre-treatment reimbursed for colon cancer patients in Spain ? (line 220 and tale 3)
- CEA ng / ml ? and cut value ?
- Nº lines of systemic therapy : why there is no TKI ?
- Ablation include ? RFA, MWA, cryotherapy ?
- mRECIST (line 261) but RECIST 1.1 (line 144), do you mean enrolled by RECIST ? but evaluate by mRECIST after Y90 or DEBIRI ? This a major discrepancy.
- Table 9: we want to know the total costs in comparision (rather than references 24 and 29 as line 416)
- Why DEBIRI group needs ICU admission, For most patients, ICU admission is not necessary after DEBIRI.
- quality of life (line 358) = there is no mention about treatment related side effects
- The conclusion is muzziness for the readers to choose Y90 or DEBIRI, finally it depends on the sources that the readers have.
Author Response
Reviewer 3:
1. Table 3. There are many space to show whole typing of abbreviations over the left colum, or the abbreviations should be presented below the table, it is difficult to get a good idea at a glance.
Ok. Done
2. Is PET-CT pre-treatment reimbursed for colon cancer patients in Spain? (line 220 and tale 3)
Yes, It is reimbursed specially in third level Spanish hospitals.
3. CEA ng / ml ? and cut value ?
Information in table 3. Done
4. Nº lines of systemic therapy : why there is no TKI ?
A least one chemotherapy line before interventional treatment (DEBIRI or TARE)
5. Ablation include ? RFA, MWA, cryotherapy ?
This information was included in table 3.
6. mRECIST (line 261) but RECIST 1.1 (line 144), do you mean enrolled by RECIST ? but evaluate by mRECIST after Y90 or DEBIRI ? This a major discrepancy.
Both are referred to mRECIST 1.1. Correction done.
7. Table 9: we want to know the total costs in comparision (rather than references 24 and 29 as line 416)
No more information about this.
8. Why DEBIRI group needs ICU admission, For most patients, ICU admission is not necessary after DEBIRI.
Some patients needs ICU admission to pain management. This is a normal practice in some hospital in Spain.
9. quality of life (line 358) = there is no mention about treatment related side effects
Yes, I agree.
10. The conclusion is muzziness for the readers to choose Y90 or DEBIRI, finally it depends on the sources that the readers have.
This paper analyzed several variables and showed advantages-disadvantages of both technique, but we can not demonstrate superiority between techniques. Maybe in the close future TARE is going to be superior in terms of short hospitalization and no pain.